# The Effect of Dye and Pigment Concentrations on the Diameter of Melt-Electrospun Polylactic Acid Fibers

**DOI:** 10.3390/polym12102321

**Published:** 2020-10-11

**Authors:** N.K. Balakrishnan, K. Koenig, G. Seide

**Affiliations:** Brightlands Chemelot Campus, Aachen-Maastricht Institute for Biobased Materials (AMIBM), Maastricht University, Urmonderbaan 22, 6167 RD Geleen, The Netherlands; kylie.konig@maastrichtuniversity.nl (K.K.); gunnar.seide@maastrichtuniversity.nl (G.S.)

**Keywords:** melt electrospinning, PLA, dope dyeing, biobased composites, nanotechnology, environmental sustainability

## Abstract

Sub-microfibers and nanofibers produce more breathable fabrics than coarse fibers and are therefore widely used in the textiles industry. They are prepared by electrospinning using a polymer solution or melt. Solution electrospinning produces finer fibers but requires toxic solvents. Melt electrospinning is more environmentally friendly, but is also technically challenging due to the low electrical conductivity and high viscosity of the polymer melt. Here we describe the use of colorants as additives to improve the electrical conductivity of polylactic acid (PLA). The addition of colorants increased the viscosity of the melt by >100%, but reduced the electrical resistance by >80% compared to pure PLA (5 GΩ). The lowest electrical resistance of 50 MΩ was achieved using a composite containing 3% (w/w) indigo. However, the thinnest fibers (52.5 µm, 53% thinner than pure PLA fibers) were obtained by adding 1% (w/w) alizarin. Scanning electron microscopy revealed that fibers containing indigo featured polymer aggregates that inhibited electrical conductivity, and thus increased the fiber diameter. With further improvements to avoid aggregation, the proposed melt electrospinning process could complement or even replace industrial solution electrospinning and dyeing.

## 1. Introduction

Electrospinning is a widely-used method for the production of microscale and nanoscale fibers, with multiple applications in the field of nanotechnology [1,2,3]. Such fibers are advantageous due to their flexibility and large surface area, making them suitable for medical applications [4,5,6,7,8], water purification [9,10], and the manufacture of electronic components [11,12] and textiles [13,14,15]. Sub-microfibers are particularly useful in textile applications because of their superior waterproofing properties and breathability compared to thicker fibers [16].

The two main forms of electrospinning are solution electrospinning, in which fibers are produced by evaporating a solvent, and melt electrospinning, in which fibers are drawn from a polymer melt [15]. Both processes are based on the same principle, in which a potential difference is established between the end of a needle capillary and a collector. This creates an electric field that induces a surface charge on the polymer solution or melt, deforming a spherical hanging droplet into a conical shape. When the electrostatic repulsive force of the surface charges overcomes the surface tension, a charged liquid jet is ejected from the tip of the Taylor cone, and the charge density on the jet interacts with the external field to create an instability that stretches the fiber, allowing it to be deposited on the collector as a nonwoven fabric [1,14,15].

Solution electrospinning is currently more economical, and thus a more widespread method for the production of nanofibers and sub-microfibers, because the lower viscosity and higher temperature of polymer solutions (compared to polymer melts) allows finer fibers to be produced using much simpler devices [17]. However, the solvents used during this process are expensive and toxic, requiring additional recovery steps to avoid carryover, particularly in biomedical applications [1,18]. For example, the production of polylactic acid (PLA) fibers requires solvents such as chloroform, dichloromethane and *N*,*N*-dimethylformamide. For other polymers, such as polypropylene (PP), no solvents are suitable for solution spinning at room temperature [19]. The evaporation of the solvent during electrospinning leaves traces on the fiber, which causes surface roughness or even defects that compromise fiber strength. The concentration of the spinning solution is very low, and evaporation of the solvent reduces the yield and wastes energy. Furthermore, capillaries often become blocked during solution electrospinning, which affects the continuous production of fibers [15,20,21].

The challenges associated with solution electrospinning have focused the efforts of researchers attempting to develop environmentally friendly melt electrospinning processes. Multiple studies have been carried out to improve melt electrospinning technology and thus overcome the limitations caused by the high temperature, high viscosity and/or low conductivity of the polymer melt, with the aim of reducing the fiber diameter [22]. Microfibers and sub-microfibers have been produced from several polymers (including polyethylene, PP and PLA) using single-fiber melt electrospinning devices and other configurations [23,24,25,26,27,28,29]. Narrower fibers can be produced by integrating a gas stream [30] or by including additives that increase conductivity and or reduce viscosity [18,31,32,33].

Colorants (dyes and pigments) are often used as additives in the textile industry, including in the dyeing of solution electrospun nanofibers. For example, bath dyeing has been applied to recycled polyester nanofibers [16] and polyamide 66 nanofibers, the latter achieved using acid dyes at lower temperatures compared to regular fibers [34]. Furthermore, dope dyeing has been reported for polyester nanofibers, thus reducing the number of processing steps and avoiding the toxic chemicals required for bath dyeing [35]. However, colorants are not only aesthetic—they can also offer other functionalities, such as better conductivity. For example, the organic semiconductor pigment copper phthalocyanine and dyes such as alizarin have been used for the development of solar cells, but are also suitable as colorants for textiles [36,37]. The π-conjugation present in these colorants is not only responsible for their color, it also improves their electrical conductivity, and it can lead to a charge carrier mobility exceeding 1 cm^2^/Vs [38]. The electric and dielectric properties of colorants reveal that conductivity follows a quantum mechanical tunnel model at low temperatures, and a correlated barrier-hopping model at higher temperatures [39].

Most studies of colorants for textiles have involved the solvent electrospinning of unsustainable fossil-based polymers. With depleting oil resources and the threat of climate change, the focus is now shifting towards biobased polymers such as PLA [40]. We previously reported that dyes could be used as conductive additives for the melt electrospinning of PLA, but more work is required to determine the suitability of colorants as conductive additives in melt electrospinning [41]. Both PLA and the colorants have polar functional groups, so their interactions can inhibit the flow of polymer chains, leading to higher viscosity. However, the higher conductivity of the colorants can negate this effect by increasing the overall conductivity of the composite. Here, we investigated the ability of several colorants (dyes and pigments) to increase the conductivity of molten PLA during melt electrospinning. We compared two potentially biobased dyes (alizarin and indigo), three fossil-based pigments (pigment blue 15:1, pigment green 7, and pigment yellow 155) and the biobased pigment pink PR122 in terms of their effect on the diameter and color of PLA fibers manufactured using a single-nozzle melt electrospinning device. We selected colorants that have a π-conjugation in their structure as a means to increase the conductivity of the polymer melt. No azo dyes were selected, even though they are also conductive, because they are toxic and mutagenic and therefore present an environmental hazard and risk to end-users [42]. We investigated the effects of the additives on the color, viscosity, morphology, thermal properties and degradation of PLA. The use of colorants as multipurpose additives avoids the use of toxic chemicals required for solvent electrospinning and conventional bath dyeing. Our results provide the basis for environmentally friendly melt electrospinning processes for the manufacture of sub-microfibers and nanofibers.

## 2. Materials and Methods

### 2.1. Materials

The biopolymer used in the study is PLA L130, with an L-content > 99%, a glass transition temperature (T_g_) of 60 °C, and a melt flow index of 24 g/10 min at 210 °C/2.16 kg. It was purchased from Total|Corbion (Gorinchem, Netherlands). The dyes alizarin (A) and indigo (I) were purchased from Sigma-Aldrich (Zwijndrecht, Netherlands). Pigment blue 15:1 (B), pigment green 7 (G), pigment yellow 155 (Y) and biobased pigment pink PR122 (P) were provided by Clariant (Louvain-la-Neuve, Belgium). The chemical structures of the colorants are shown in Table 1.

The presence of copper ions is likely to make the inorganic pigments more conductive, but also more difficult to disperse in the polymer matrix. In contrast, although the dyes and organic pigments may have a lower conductivity, they should be easier to disperse.

### 2.2. Methods

#### 2.2.1. Microcompounder

PLA undergoes degradation by hydrolysis at high temperatures, so we avoided this by drying PLA and the plasticizer at 80 °C overnight in a vacuum before compounding. The colorants were dried at 50 °C in a vacuum for 2 h prior to compounding trials, which were carried out using a co-rotating microcompounder from Xplore (Sittard, Netherlands). We prepared three 5 g batches of each composite at 200 °C with a screw speed of 100 rpm and a mixing time of 2 min. Rheometer plates were molded in an IM 5.5 injection molding machine (Xplore). The composites and their abbreviations are listed in Table 2.

#### 2.2.2. Melt Electrospinning Equipment

We used a single-fiber melt electrospinning device, comprising a temperature controller, high-voltage power supply, heating elements, syringe pump and collector (Figure 1). The device was equipped with JCS-33A temperature process controllers (Shinko Technos, Osaka, Japan) and PT 100 platinum thermocouples (Omega Engineering, Deckenpfron, Germany) to maintain the melting temperature at 275 °C. We used a KNH65 high-voltage generator (Eltex-Elektrostatik, Weil am Rhein, Germany) to maintain a constant +50 kV on the collector (a 6 cm aluminum plate overlaid with a thin paperboard) while grinding the spinneret, which was a 2 mL glass syringe (Poulten and Graf, Wertheim, Germany) with a nozzle orifice of 1 mm, and this was placed at a distance 10 cm from the collector. An 11 Plus spin pump (Harvard Apparatus, Cambridge, MA, USA) was used with a constant delivery rate of 4 mL/h.

#### 2.2.3. Characterization of Composites

The thermal properties of the composites were analyzed by differential scanning calorimetry (DSC) and thermogravimetric analysis (TGA). For DSC, we used a Q2000 device (TA Instruments, New Castle, DE, USA) for two heating cycles between 25 and 200 °C at a rate of 10 °C/min in a nitrogen atmosphere, and analyzed the data obtained from the second cycle. We compared the T_g_, crystallization temperature (T_c_) and melting point (T_m_) from the second cycle. The percentage crystallinity (Xc) of PLA and its composites was calculated by taking 93.7 J/g as the melt enthalpy of 100% crystalline PLA [45]. The same experimental protocol was used to prepare all the composites, so any differences in the properties would depend on the additive and its quantity. The effect of additives on the thermal degradation of PLA was assumed to be most evident at the highest additive concentration, so TGA was applied to composites containing 3% (w/w) of each additive. PLA and its composites were heated from room temperature to 500 °C at a rate of 10 °C/min in a nitrogen atmosphere, and the temperatures at which 5% and 50% weight loss occurred were determined and compared.

We measured changes in PLA viscosity using a Discovery HR1 hybrid rheometer (TA Instruments, Asse, Belgium) fitted with a 25 mm plate (gap = 1000 µm) and a shear rate increasing from 0.01 to 500 1/s. We maintained the strain amplitude at 1% in a constant environment with a nitrogen atmosphere at 200 °C, and used an equilibration time of 5 min. For more relatability, we compared the viscosity of PLA and its composites at a shear rate of 5 1/s.

A 1260 Infinity gel permeation chromatography (GPC) system (Agilent Technologies, Santa Clara, USA) was used to measure molecular degradation in pure PLA and its composites. We used hexafluoro-2-isopropanol (HFIP) containing 0.19% sodium trifluoroacetate as the solvent and mobile phase. We prepared solutions containing 5 mg PLA and its composites, passed them through a polytetrafluoroethylene filter, and injected them into a silica column. Calibration was performed using a standard polymethyl methacrylate polymer (1.0 × 10^5^ g/mol), and we compared the weight average molecular weight (M_w_), number average molecular weight (M_n_) and polydispersity index (PDI).

The electrical resistance of PLA and its composites was measured at 325 °C using a Keithley 617 electrometer (Tektronix Inc., Beaverton, OR, USA), as shown Figure 2. The polymer granulate was melted using band heaters, and two electrodes (6 mm apart) were dipped in the melt and connected to the electrometer. The electrical current flowing between the electrodes was measured by applying a constant 10 V.

#### 2.2.4. Characterization of Fibers

We measured fiber diameters by reflected light microscopy using a DM4000 M instrument (Leica Microsystems, Wetzlar, Germany) at 100–200× magnification, and images were captured using Leica Application Suite software. In total, 10 images representing different areas of each nonwoven were used to determine the average fiber diameter.

The dispersion of the colorant was determined by scanning electron microscopy (Leo 1450 VP from Zeiss, Oberkochen, Germany) at an acceleration tension of 15 kV. The samples were freeze-fractured after immersion in liquid nitrogen and then sputtered with gold and mounted on SEM stubs. Cross-section and surface images were acquired in secondary electron mode and at different magnifications depending on the amount of additive.

Color measurements were collected by spectrophotometry in the range 400–800 nm according to ISO 11664-4:2008 using a UV3600 spectrophotometer device (Shimadzu, Kyoto, Japan). A 50 W halogen lamp was used as the light source, and a photomultiplier tube and PbS were used as the detector. The spectrum was recorded in the reflectance mode after white and black calibrations. Spectrophotometric data were transferred from the instrument to CIELab software to define the color coordinates of each sample.

#### 2.2.5. Experimental Methodology

Our experimental methodology is summarized in Figure 3. We compounded PLA with the additives first, and divided the resulting composites into two parts. The first was injection molded into plates and used for rheological analysis. The second was cut into small granules for characterization and melt electrospinning. Our goal was to reduce the diameter of the fibers by increasing the electrical conductivity and/or reducing the viscosity of the melt. The other process parameters (e.g., temperature, electrical field, and throughput) were kept constant so that any change in fiber diameter could be attributed to the type and quantity of additive.

## 3. Results and Discussion

### 3.1. Thermal Properties of the PLA Composites

The DSC thermograms of PLA and the composites containing 2% (w/w) of each additive are presented in Figure 4. These are representative results because the other composites followed the same trend, and are therefore included in Appendix A (Figure A1 and Figure A2).

The DSC thermograms indicate the T_g_, T_c_ and T_m_. In most cases, we observed a single melting peak during the heating phase, although the composites with alizarin, indigo and pink pigment featured a doublet around the same T_m_. This may reflect the formation of imperfect crystals during the first cooling phase, which melt and fuse together to form perfect crystals, and then melt again at the higher temperature. A similar phenomenon was reported for the PLA and carbon black composites [46]. The T_g_, T_c_ and Tm of PLA and its composites are summarized in Figure 5.

The DSC results showed that the T_g_ of PLA is not markedly affected by the nature or concentration of the additive, remaining at ~60 °C in agreement with previous studies [47]. Similarly, the Tm remained at ~170 °C, and the Xc at ~50% (as expected, given that the second heating/cooling cycle was considered and the thermal history of all the PLA composites was therefore the same). In contrast, we observed a significant additive-dependent difference in the T_c_, which was 104 °C for pure PLA and composites containing the green and pink pigments. However, The Tc increased by more than 20 °C in the composites containing alizarin or the blue and yellow pigments, reaching maximum values of 129 °C (A1), 134 °C (B2) and 131 °C (Y1), respectively. Interestingly, although the Tc increased linearly with the weight percentage of indigo, there was a plateauing effect for the yellow pigment and no clear relationship between Tc and weight percentage for the other colorants (Figure 5, black rings). These results suggest that alizarin, indigo, and the blue and yellow pigments have a nucleating effect on PLA, a phenomenon also extensively observed for PP and colorants, including the blue pigment tested here [48]. The nucleating effect in PP reflects the fact that the blue pigment exists in different polymorphic forms, providing surfaces that match PP chains and thus enabling the epitaxial growth of PP crystals [49]. It is possible that the nucleation in PLA involves a similar mechanism, but this needs to be investigated in more detail to determine the exact mechanism.

The TGA thermograms of PLA and composites containing 3% (w/w) of each additive are presented in Figure 6 and the temperatures at which 5% and 50% weight loss occurred are compared in Figure 7. For pure PLA, we found that a 5% weight loss occurs at 350 °C and a 50% weight loss occurs at 378.0 °C, which is consistent with previous reports [50]. The lowest 5% weight loss temperature was 342 °C for composite G3, and the highest was 352 °C for composite A3. The lowest 50% weight loss temperature was 375 °C for composite I3 and the highest was 382 °C for composite A3. There was no significant difference in the degradation temperatures of PLA and composites containing 3% (w/w) of each additive.

### 3.2. Effect of Additives on Melt Viscosity

The rheogram of PLA and its composites containing 3% (w/w) of each additive is shown in Figure 8. Since the rheogram of composites containing other weight% of additives was similar to this, it is presented in Appendix A (Figure A3 and Figure A4). All the samples exhibited non-Newtonian behavior (shear thinning). The viscosity was very high at low shear rates, and fell sharply with increasing shear rate. Although the viscosity increased following the incorporation of each additive, in most cases the composites showed similar rheological behavior to pure PLA. The increase in viscosity was the highest for the pink pigment.

The incorporation of colorants significantly increased the viscosity of PLA, especially at the lower shear rate of 5 1/s (Figure 9). 

Similarly, the viscosity of PLA was shown to increase when mixed with carbon black pigment [51]. Given that both PLA and the additives contain polar groups, it is likely that hydrogen bonds forming within the mixture inhibit the slippage of PLA chains [52]. This is more evident at lower shear rates, because the shear forces are not strong enough to break these interactions, leading to lower chain mobility and thus higher viscosity.

The viscosity of pure PLA at a shear rate of 5 1/s is 120 Pa·s. Adding 1% (w/w) alizarin increases the viscosity by >100% to 261 Pa·s, but the effect of increasing the additive concentration is less significant. The viscosity of A3 (328 Pa·s) is only 25% higher than that of A1. A similar trend was observed for the other composites, indicating that the type of additive affects the viscosity more than the concentration, as previously reported for PP composites with pigments [53].

The diameter of melt electrospun fibers can be reduced either by increasing the conductivity or reducing the viscosity of the polymer [54]. Because all the additives caused a significant increase in viscosity, any reduction in fiber diameter achieved by the PLA composites can only be attributed to an increase in electrical conductivity.

### 3.3. GPC Analysis of PLA and Its Composites

The relative Mw of extruded PLA was 150,940, the Mn was 96,977, and the PDI was 1.56. These values did not change significantly for the composites. The lowest Mw and Mn values were observed for P3 (148,860 and 92,850, respectively) and the highest were observed for G2 (167,900 and 103,760, respectively) as shown in Figure 10. The PDI for all composites varied between 1.55 and 1.65 (Figure 11). There was no significant change in any of these values caused by the additives, suggesting that none of the colorants induce the degradation of PLA.

### 3.4. Analysis of Additive Dispersion by SEM

The dispersion of the colorants was investigated by SEM. Images of PLA composite fibers containing 3% (w/w) of each additive are shown in Figure 12. No aggregates were observed in the A3 and P3 fibers, but the distribution of alizarin was more uniform whereas the pink pigment was not uniformly dispersed. The absence of aggregates suggests that the intermolecular interactions between PLA and these colorants are stronger than the intramolecular interactions between the colorant particles or molecules. Small colorant aggregates were observed on the Y3 fibers, but much larger aggregates, some up to 10 µm in diameter, were observed on the B3, G3 and I3 fibers. Accordingly, the intramolecular interactions between the colorant particles or molecules must be stronger in these fibers than the intermolecular interactions between the colorants and PLA. Aggregate formation is not favorable for electrospinning, because the electrical conductivity is enhanced by well-dispersed additives, allowing them to form a conductive network [55]. In our electrospinning setup (Figure 2), we use a glass syringe with a plunger for spinning. The composite is filled and melted inside the syringe, and the blue, green and yellow pigments, as well as indigo dye, could aggregate in the melt because no shear is applied and their movement is unrestricted. A possible optimization step would be the integration of a screw system to provide shear and thus improve the dispersion.

### 3.5. Effect of Additives on Melt Conductivity

The electrical resistance of pure PLA and its composites was measured at a higher temperature than the spinning process (325 °C) to account for the heat energy lost over the larger surface of the beaker. Electrical conductivity requires mobile charge carriers, so the electrical resistance of pure PLA (5.0 GΩ) decreased at least by a factor of five in the presence of any of the colorants (Figure 13).

For all composites, the electrical resistance decreased with the increasing additive concentration. The most significant decrease (~99%) was observed for the PLA/indigo composites. Alizarin and the pink and yellow pigments had similar moderate effects on electrical resistance at the lowest concentration of 1% (w/w), but alizarin had a stronger effect than either of the pigments at higher concentrations. The blue and green pigments had a stronger effect than alizarin and the pink and yellow pigments. In agreement with the SEM data, colorants that favor aggregate formation (indigo and the blue and green pigments) formed composites in which electrical resistance is less dependent on the additive concentration. The dispersion is poor at higher concentrations, and thus the additive’s influence on electrical resistance is lower than expected. SEM showed that alizarin and the pink pigment were well dispersed, which can lead to the formation of a conductive network and thus lower electrical resistance with increasing concentration [56].

### 3.6. Fiber Diameters Achieved Using Different PLA Composites

Next, we used our single-fiber melt electrospinning device to evaluate the processability of the PLA composites and determine how the colorants affected the fiber diameter. Taylor cone formation followed by typical fiber deposition was successful for pure PLA and all the composites at a temperature of 275 °C. The average fiber diameter of pure PLA (112.5 µm) was reduced by all the additives we tested (Figure 14).

As stated above, narrower fibers can be produced by melt electrospinning if the viscosity of the melt is reduced and/or the electrical conductivity increased. Given that all additives increased the viscosity of the melt, and viscosity was driven more by the type of colorant than the weight percentage, it was apparent that the reduction in fiber diameter must reflect an increase in electrical conductivity. The finest fibers (52.5 µm, 53% narrower than pure PLA) were achieved for composite A1. However, even though the conductivity of the PLA/alizarin composites increased at higher alizarin concentrations, there was no significant difference in the diameters of the A1 and A3 fibers. Furthermore, the A1 and A3 fibers were also similar in thickness to the B2, G3, I3 and P3 fibers. Composites containing the pink pigment were more viscous than those containing alizarin, whereas the electrical conductivity was almost the same, hence the fibers containing alizarin were marginally narrower. The B2, G3 and I3 fibers were among the finest despite the poor dispersion of the additives observed by SEM, suggesting the additives have higher conductivity than PLA. This means it should be possible to reduce the fiber diameter even further by improving the dispersion of additives using a melting unit that generates shear. Although the electrical conductivity of indigo-containing composites was the highest when measured in the beaker, this did not lead to the narrowest fibers, contrary to our expectations. We assume that the aggregation behavior in the beaker differs from that in the spinning device. Due to the greater space available in the beaker, colorant aggregation could lead to the formation of a network on the base of the vessel, making it easier for the applied load to flow. This possibility should be addressed in future investigations.

### 3.7. Color Measurement of Electrospun Fibers

The colorimetric analysis of fibers obtained from PLA and its composites is presented in Table 3. In the CIE-Lab color measurement system, L* represents lightness (L* = 100 for a perfectly reflective surface (white) and L* = 0 for a perfect black surface). Redness is represented by a* > 0, whereas greenness is represented by a* < 0. Similarly, b* > 0 represents yellowness, and b* < 0 represents blueness [57]. The presence of colorants caused the lightness index to decline compared to pure PLA, and in most cases the lightness index fell further as the additive concentration increased. The exception was the yellow pigment, where increasing the concentration caused both L* and b* to increase, indicating the formation of a brighter yellow shade.

## 4. Conclusions and Further Perspectives

We have investigated the effect of colorants (dyes and pigments) as conductive additives in the melt electrospinning of PLA, specifically their impact on fiber diameter. We produced electrospun fibers with diameters in the micrometer range. This environmentally friendly process could potentially replace conventional solution electrospinning to produce microfibers and nanofibers, which are used in textiles because of their superior breathability. This process is also advantageous because it allows inline dyeing to make colored textiles. All composites led to the formation of a Taylor cone in the melt electrospinning equipment, which was followed by the deposition of fibers on the collector plate. The diameter of the pure PLA fibers was >100 µm, but this was reduced by at least 50% following the addition of all colorants except the yellow pigment. The best result was achieved with composite A1, which formed fibers 52.5 µm in diameter (53% narrower than pure PLA). The composites containing alizarin dye and the pink pigment had similar electrical resistance, but the viscosity of composites containing the pink pigment was approximately three times higher than pure PLA, producing slightly thicker fibers. The composite containing yellow pigment showed similar electrical conductivity, but SEM analysis revealed the formation of small aggregates, which were not present in composites containing alizarin dye or the pink pigment. The electrical conductivity of a composite is inhibited by the non-homogenous dispersion of the additive, which probably explains the thickness of fibers containing the yellow pigment. Composites containing the indigo dye, blue pigment and green pigment had a lower electrical resistance than pure PLA—99% lower in the case of composites containing indigo dye. However, the diameter of the fibers was not as low as anticipated, probably due to the formation of large aggregates. Melt electrospinning is carried out using a syringe with a plunger, whereas electrical conductivity is tested in a beaker with a wide base, probably leading to different aggregation behaviors. The greater amount of space in the beaker may promote the formation of a conductive network that confers lower electrical resistance, which is not likely to occur in the melt electrospinning equipment. Taken together, our results suggest that dyes and pigments make useful functional additives, but their impact on fiber diameter must be evaluated on a case by case basis, to account for the conflicting effects on viscosity, conductivity and aggregate formation.

## Figures and Tables

**Figure 1 polymers-12-02321-f001:**
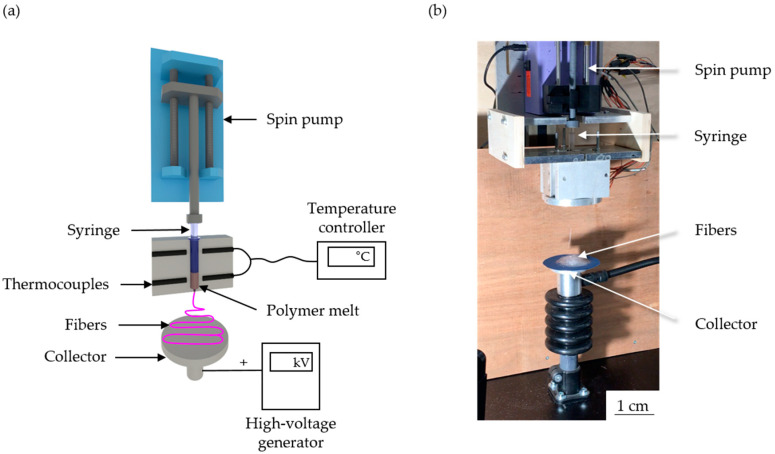
Schematic illustration (**a**) and picture (**b**) of our melt electrospinning equipment.

**Figure 2 polymers-12-02321-f002:**
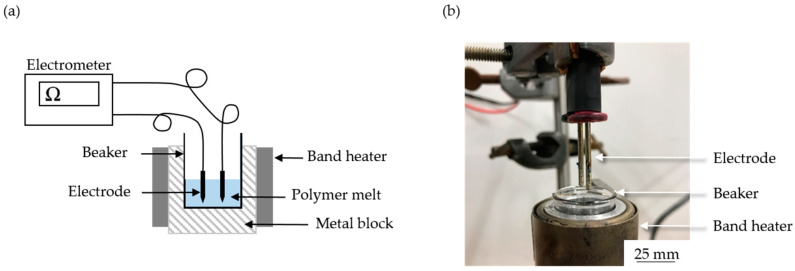
Schematic illustration (**a**) and picture (**b**) of our electrometer setup.

**Figure 3 polymers-12-02321-f003:**
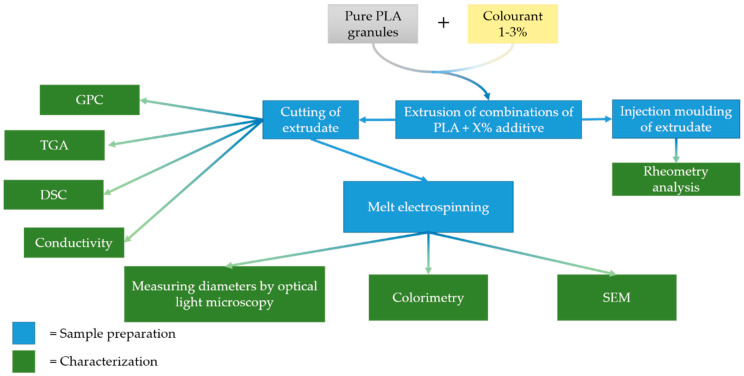
Methodology for the preparation and analysis of electrospun PLA fibers.

**Figure 4 polymers-12-02321-f004:**
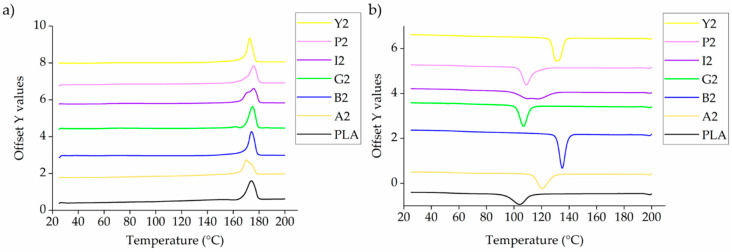
DSC thermograms of PLA and composites containing 2% (w/w) of each additive during (**a**) the heating phase and (**b**) the cooling phase.

**Figure 5 polymers-12-02321-f005:**
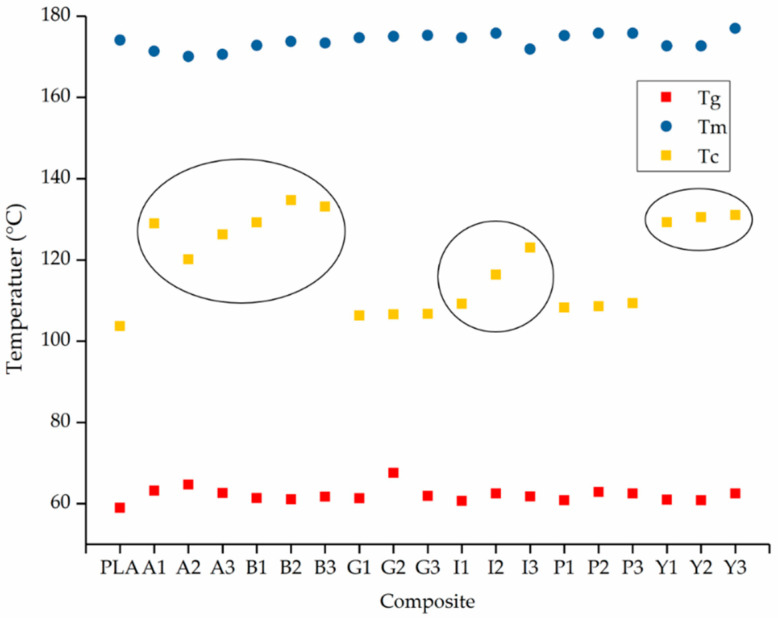
The T_g_, T_c_ and T_m_ of PLA and its composites. The three rings indicate the relationship between T_m_ and weight percentage for alizarin, indigo, and the blue and yellow pigments. (The composites having a different T_C_ have been circled for easy identification)

**Figure 6 polymers-12-02321-f006:**
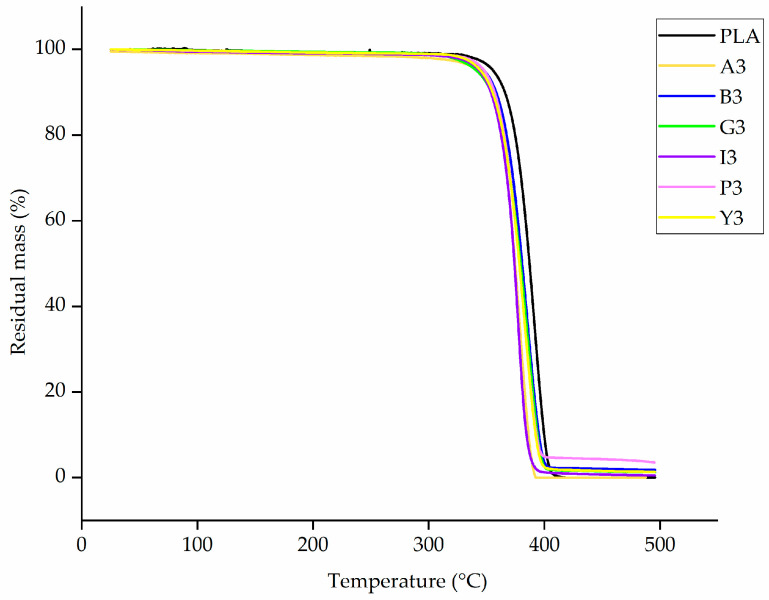
TGA thermogram of PLA and its composites containing 3% (w/w) of each additive, showing no significant difference between the degradation curves.

**Figure 7 polymers-12-02321-f007:**
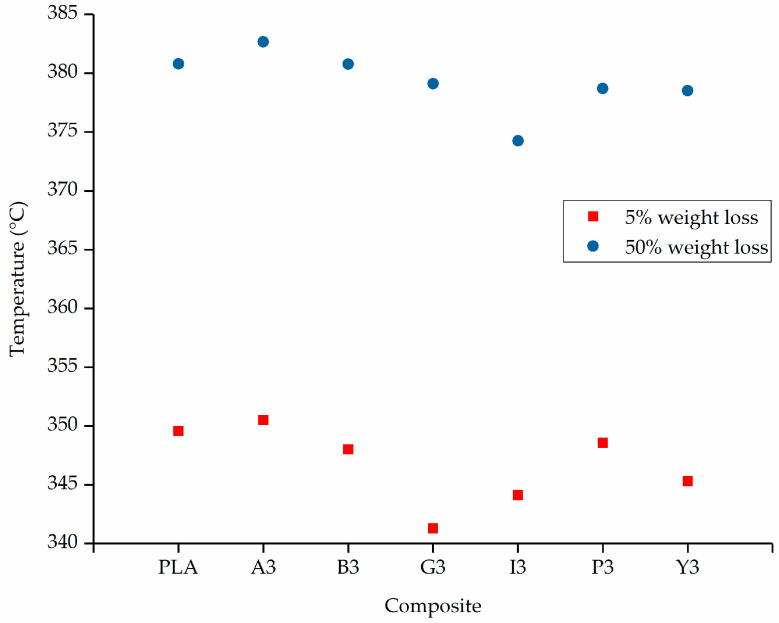
Comparison of 5% and 50% weight loss temperatures for PLA and composites containing 3% (w/w) of each additive.

**Figure 8 polymers-12-02321-f008:**
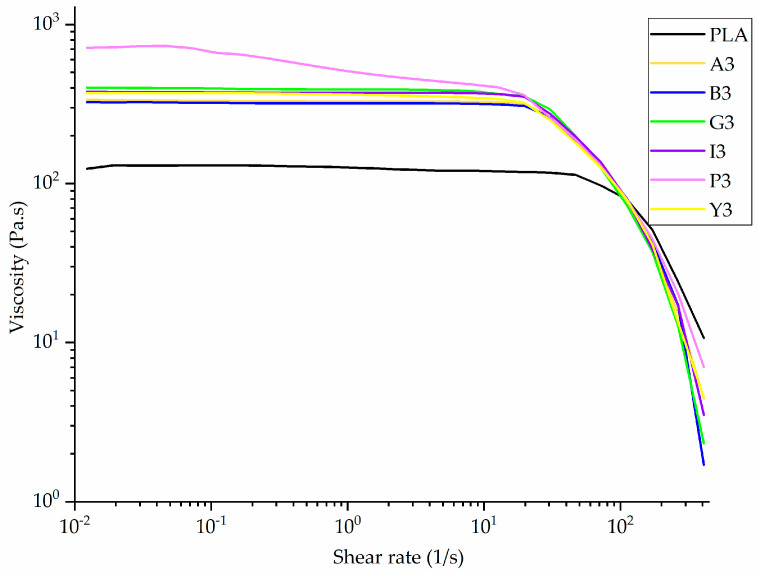
Rheogram of PLA and its composites containing 3% (w/w) of each additive.

**Figure 9 polymers-12-02321-f009:**
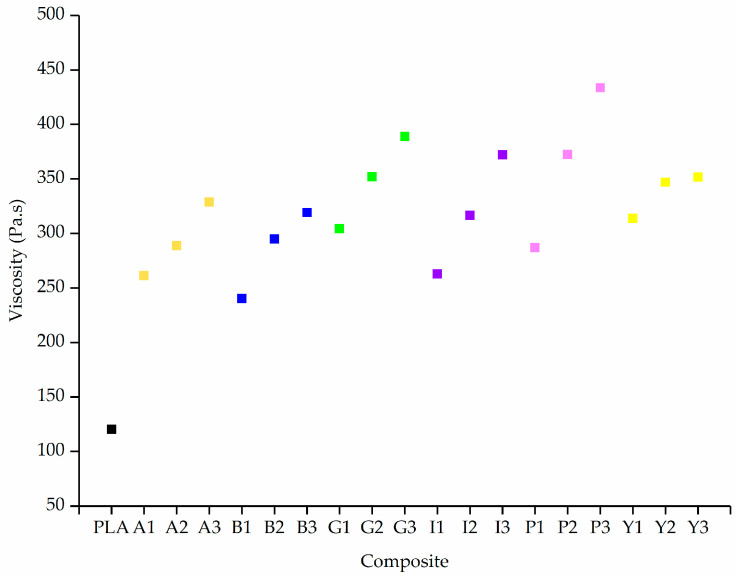
Viscosity of PLA and its composites containing 1%, 2% and 3% (w/w) of each additive at a shear rate of 5 rad/s. The colored squares indicate the different dyes/pigments for visual clarity.

**Figure 10 polymers-12-02321-f010:**
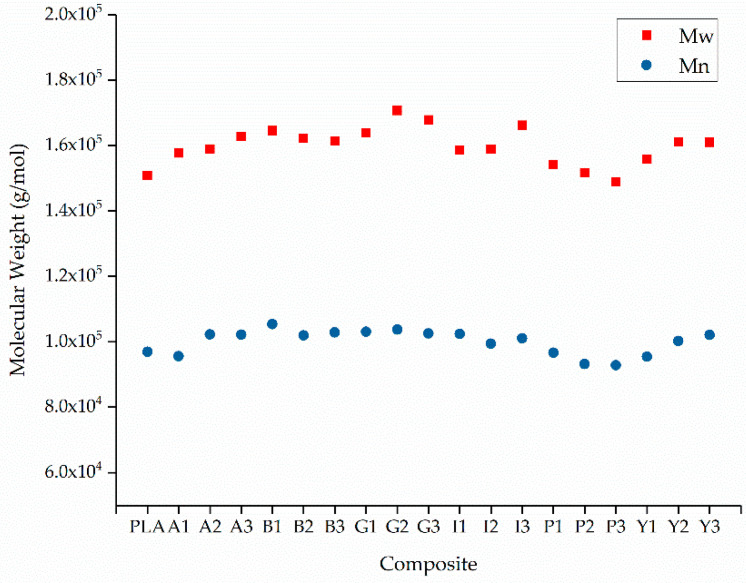
The weight average molecular weight (Mw) and number average molecular weight (Mn) of PLA and its composites containing 1%, 2% and 3% (w/w) of each additive.

**Figure 11 polymers-12-02321-f011:**
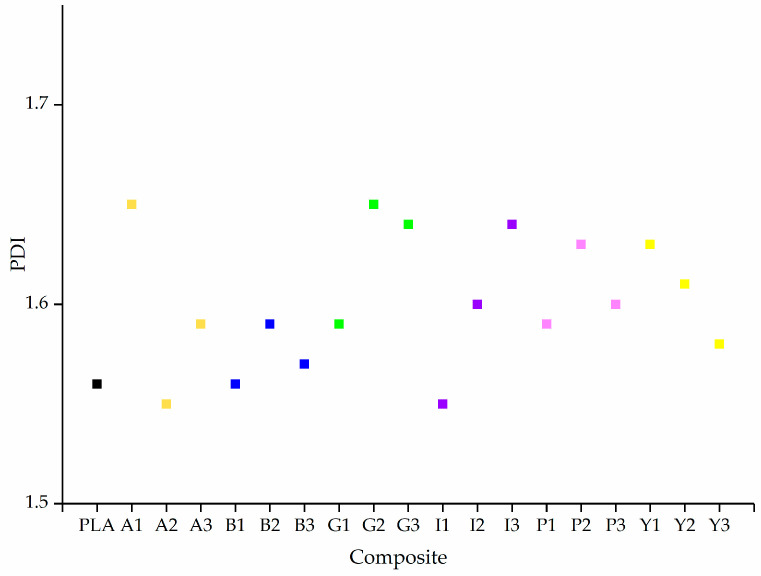
The polydispersity indices (PDI) of PLA and its composites containing 1%, 2% and 3% (w/w) of each additive. The colored squares indicate the different dyes/pigments for visual clarity.

**Figure 12 polymers-12-02321-f012:**
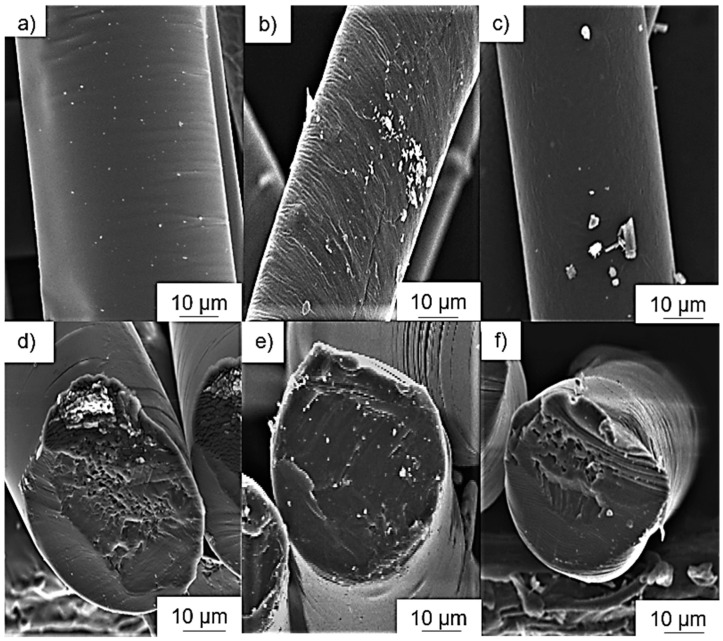
Scanning electron micrographs of PLA composite fibers: (**a**) A3, (**b**) B3, (**c**) G3, (**d**) I3, (**e**) P3, and (**f**) Y3.

**Figure 13 polymers-12-02321-f013:**
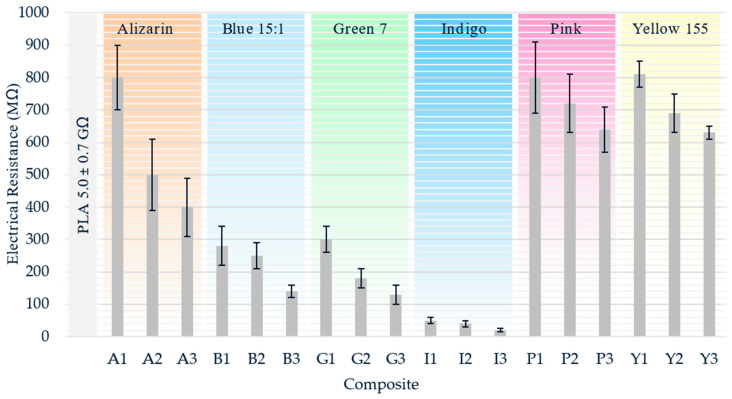
Electrical resistance of PLA and its composites containing 1%, 2% and 3% (w/w) of each additive.

**Figure 14 polymers-12-02321-f014:**
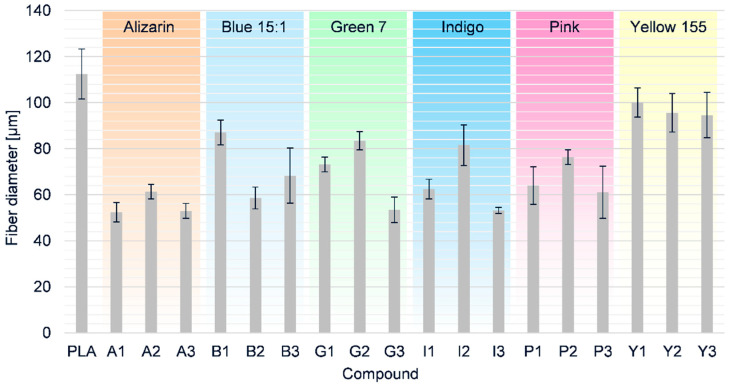
Fiber diameters and standard deviations of PLA and its composites containing 1%, 2% and 3% (w/w) of each additive produced by melt electrospinning at 275 °C using a single-nozzle laboratory device.

**Table 1 polymers-12-02321-t001:** Chemical structures and melting points of the colorants used in this study [43,44]

Additive	Chemical Structure	Melting Point (°C)
Alizarin	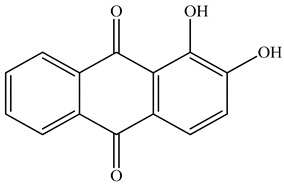	279–283
Blue pigment 15:1	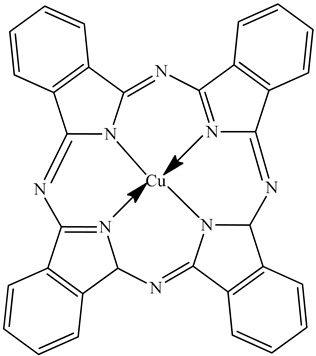	350
Green pigment 7	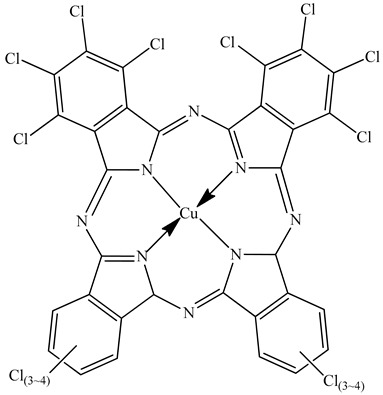	480
Indigo	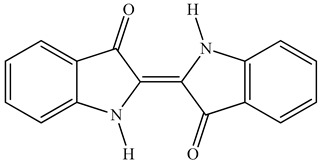	>300
Pink pigment PR122	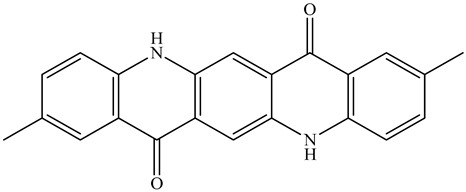	Not available
Yellow pigment 155	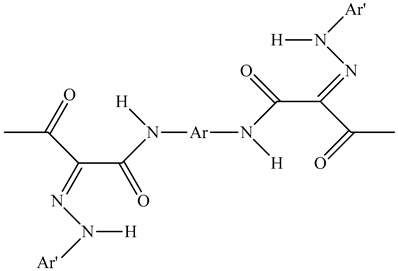	>280

**Table 2 polymers-12-02321-t002:** List of the 18 composites produced for this study. The composite names were based on the initial letter of each additive and the weight percentage of the additive.

Composite Abbreviation	Colorant Name	% (w/w) of Colorant
A1	Alizarin dye	1
A2		2
A3		3
B1	Blue pigment	1
B2		2
B3		3
G1	Green pigment	1
G2		2
G3		3
I1	Indigo dye	1
I2		2
I3		3
P1	Pink pigment	1
P2		2
P3		3
Y1	Yellow pigment	1
Y2		2
Y3		3

**Table 3 polymers-12-02321-t003:** Colorimetric values of PLA and its composites containing 1%, 2% and 3% (w/w) of each additive.

Composite	L*	A*	B*
PLA	67.22	−0.25	2.06
A1	40.42	−7.94	36.22
A2	37.69	−3.58	39.05
A3	31.42	−1.46	32.79
B1	19.45	6.01	−27.8
B2	16.95	5.14	−23.09
B3	13.53	7.5	−20.53
G1	36.27	−31.88	−0.42
G2	32.61	−32.28	−0.97
G3	25.16	−22.43	−1.84
I1	29.7	0.3	−15.54
I2	25.58	4.42	−22.78
I3	23.96	4.41	−20.86
P1	27.09	43.75	−5.42
P2	24.03	39.14	3.01
P3	22.35	36.95	2.92
Y1	50.9	−15.03	49.66
Y2	54.82	−9.52	61.13

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
