# Peer review of "The Effect of Dye and Pigment Concentrations on the Diameter of Melt-Electrospun Polylactic Acid Fibers"

_polymers, 2020, doi:10.3390/polym12102321_

Round 1
Reviewer 1 Report
The paper introduces the use of dyes to control viscosity and conductivity during melt electrospinning. From the environmental perspective, this innovation is better than the solution alternative as its eliminates toxic solvents. The author should consider the following points in their revision. The authors should provide the rationale for selecting the dyes. Does these dyes has special functionalities and structure that the author hypothesize to control viscosity and conductivity? As noted by the authors, the effect of the dyes on viscosity and conductivity is conflicting and the author should explain the origin of this conflict. From the environmental perspective, are these dyes safer than the solvents?
Author Response
Dear reviewer,
Thank you very much for the much appreciated and constructive feedback. We have made some changes to address the remarks and we hope we have sufficiently answered the questions. If you have any other questions, kindly let us know.
Thanks in advance for considering our manuscript.
Best regards,
Naveen

Reviewer 2 Report
The manuscript presents a study on the impact of various dyes on the properties and morphology of nanofibres. Overall the paper is poorly drafted and organized. it is very difficult to follow up the discussion and the different samples. The naming is awful and the graphs are often not plotted in any meaningful way. Per se, it is not possible to recommend publication.
There are lots of studies designed around the systematic control of nanofibres diameters and properties. Most have looked at the incorporation of various additives to control such properties. Besides changing polymer concentration and spinning conditions which were extensively investigated obviously there are also heaps of materials with systematic studies on such engineering. The purpose of the discussion and the benchmarking is poor and should be reinforced.
Here it is truly difficult to correlate some measured properties to the performance. Most figures are not organized around any logical dynamic. They are listing samples but not ordering them or plotting data against other relevant data. Plotting is poor - one should not have the chart bars with the values and st dev on top... ordering of the series is so unclear.
The series of samples (for instance content in dye per type of samples) are not well defined around the properties of these individual dyes. Variations of properties could be related to the native dyes obviously but this was not done clearly. Last some properties evaluated go way beyond that of the topic/abstract. Some modelling (DFT maybe) should help explain the trends.
Altogether - this is a poorly prepared report which requires extensive work to be publishable. Novelty is limited and so is significance.
Author Response
Dear reviewer,
Thank you for the comments. We have made our best attempt to work on the comments. Please find attached the changes made. For the remaining comments, we request you to kindly refer to the letter to the editor.
Best regards,
Naveen
